# An Imidazo[1,5-a]pyridine Benzopyrylium-Based NIR Fluorescent Probe with Ultra-Large Stokes Shifts for Monitoring SO_2_

**DOI:** 10.3390/molecules28020515

**Published:** 2023-01-05

**Authors:** Renle Cui, Caihong Liu, Ping Zhang, Kun Qin, Yanqing Ge

**Affiliations:** Department of Chemistry and Pharmaceutical Engineering, Shandong First Medical University & Shandong Academy of Medical Sciences, No. 619, Changcheng Road, Taian 271016, Chinawangrui@sdfmu.edu.cn (K.Q.)

**Keywords:** ultra-large Stokes shifts, imidazo[1,2-a]pyridine, bioimaging, FRET, mitochondria

## Abstract

A mitochondria-targeted NIR probe based on the FRET mechanism was developed. It shows ultra-large Stokes shifts (460 nm) and emission shifts (285 nm). Furthermore, we also realized the imaging of SO_2_ in living SKOV-3 cells, zebrafish and living mice which may be useful for understanding the biological roles of SO_2_ in mitochondria and in vivo.

## 1. Introduction

Sulfur dioxide, a well-known atmospheric pollutant, has been regarded as a new possible gas transmitter following NO, CO and H_2_S [1,2,3,4]. It plays important roles in many physiological processes. SO_2_ can dissolve easily in water to form its derivatives bisulfite (HSO_3_^−^) and sulfite (SO_3_^2−^), so the physiological functions of SO_2_ can be attributed to its derivatives (HSO_3_^−^/SO_3_^2−^). However, a high level of endogenous SO_2_, generated by the oxidation of H_2_S and thiol-containing amino acids in mitochondria, may bring about neurological disorders, cancers and other diseases [5,6,7,8]. Hence, it is greatly important to establish sensitive and rapid methods for SO_2_ detection to further gain insight into its functions in biological systems, especially in mitochondria.

Recently, fluorescent probes have become a powerful tool in biological imaging owing to their simplicity, high selectivity and small cell damage [9,10,11,12]. Different from traditional intensity-based probes, ratiometric probes are independent of the probe concentration, environment and excitation intensity [13,14,15]. Besides the ICT (Intramiolecular Charge Transfer)-based ratiometric probes, fluorescence resonance energy transfer (FRET)-based ratiometric probes are the most widely designed and used (Appendix A). Until now, numerous FRET-based SO_2_ probes have been designed and synthesized due to their large pseudo-Stokes shifts, avoiding interference of a biological background [16,17,18,19,20,21,22,23,24,25].

As classic fluorophores, hemicyanines have drawn increasing attention because of their simple synthesis and excellent response to SO_2_ [26]. Their derivatives were selected as acceptors to construct FRET probes [27,28]. However, the emission of the hemicyanines is around 600 nm, which seriously limits their application in vivo. Therefore, it is of significance to search for new fluorophores, especially with NIR emission, as acceptors.

On the other hand, to build an effective FRET platform, the development of new fluorophores as donors whose emission overlaps well with the absorption of acceptors is essential. Owing to the good optical properties [29], imidazole[1,5-a]pyridines were selected as the donor to construct the FRET platform [30]. In addition, we chose benzopyran salt as the acceptor because of its NIR emission. Meanwhile, the benzopyran moiety could not only be used as a reactive site for the Michael addition reaction with SO_2_ to achieve detection purposes, but it could also target mitochondria due to positive electricity. Therefore, the designed probe IPB-RL-1 could successfully achieve its imaging of SO_2_ in mitochondria in SKOV-3 cells.

## 2. Results and Discussion

### 2.1. Synthesis of IPB-RL-1

The probe IPB-RL-1 was easily prepared using a classic organic reaction, as shown in Figure 1. The structure was confirmed by NMR and HRMS.

### 2.2. Optical Properties of IPB-RL-1

To examine the optical properties of **IPB-RL-1**, we first examined its selectivity. As shown in Figure 1a,b, there were no obvious changes in absorption and emission after the probe reacted with various ions (Br^−^, CH_3_COO^−^, Cl^−^, ClO_4_^−^, ClO^−^, F^−^, H_2_PO_4_^−^, HCO_3_^−^, HPO_4_^2−^, HS^−^, I^−^, NO_2_^−^, NO_3_^−^, S_2_O_8_^2−^, SO_4_^2−^, GSH, and Cys). However, when SO_3_^2−^ was added, it was clearly observed by the naked eye that the probe solution changed from blue to colorless, and the fluorescence intensity was quenched at 760 nm, indicating that **IPB-RL-1** showed good selectivity for the detection of SO_3_^2−^. The anti-interference experiment (Appendix A) demonstrated that **IPB-RL-1** had good anti-interference performance and could specifically detect SO_3_^2−^ even in the presence of other ions.

For a better application in living systems, UV–vis and fluorescence titration experiments were also carried out. As shown in Figure 2a, **IPB-RL-1** has a strong UV–vis absorption peak at 620 nm in the solution of DMSO/PBS (V/V = 3/7). Yet, with the continuous addition of SO_3_^2−^, the absorption peak at 620 nm decreased and the absorption peak at 310 nm increased. Meanwhile, the naked eye captured a rapid color change of the probe solution from blue to colorless. The near-infrared fluorescence emission peak at 760 nm decreased with the increase of SO_3_^2−^ while the emission peak at 470 nm increased (Figure 2b), which further confirmed that the FRET was turned off. In addition, an excellent linear correlation between the ratio F_470_/F_760_ and SO_3_^2−^ concentration was observed. The detection limit was calculated to be 0.98 μM using the linear regression curve (Appendix A) and LOD formula (LOD = 3 σ/k, σ is the standard deviation of the blank measurement, and k is the slope of the fluorescence emission ratio (I_475_/I_760_) and SO_3_^2−^ concentration). In the process of monitoring the reaction time between **IPB-RL-1** and SO_3_^2−^, the fluorescence intensity reached equilibrium (Appendix A) in a very short time (less than 10 s). These results indicated that **IPB-RL-1** was suitable for further application in imaging in cells and in vivo.

The results of the MTT (Methyl Thiazolyl Tetrazolium) experiment (Appendix A) showed that **IPB-RL-1** had a lower cytotoxicity to SKOV-3 cells and could be used for further cell imaging experiments. In Figure 3, fluorescence in the red and blue channels were observed after SKOV-3 cells were incubated with the probe for 1 h. However, when the cells were incubated with the probe for 1 h and then incubated with SO_3_^2−^ for 20 min, the fluorescence in the blue channel was enhanced and the fluorescence in the red channel was significantly weakened, which suggested that probe **IPB-RL-1** could be used to detect SO_3_^2−^ in SKOV-3 cells.

Next, since the benzopyran part of **IPB-RL-1** is positively charged, the mitochondria-targeted experiment was tested. As shown in Figure 4, the red fluorescence of MitoTracker Red and the blue fluorescence of probe **IPB-RL-1** overlap well (coefficient = 0.91).

Owning to the excellent properties of **IPB-RL-1** in cell imaging, its capability for the visualization of SO_3_^2−^ in zebrafish was examined. As depicted in Figure 5, weak blue and red fluorescent signals were observed in the control group. When the zebrafish were incubated with **IPB-RL-1** for 1 h, the fluorescent signals became obviously strong both in the blue channel and the red channel. Yet, when the zebrafish were incubated with **IPB-RL-1** for 1 h and then Na_2_SO_3_ for 30 min, the fluorescent signals in the red channel became obviously weak while there was no significant change in the blue channel. Therefore, we believe that **IPB-RL-1** can effectively image in vivo. Hence, imaging in mice was conducted to further explore its application advantages. As NIR fluorescence emission is required for the experiments in vivo, only fluorescence changes in the 698–766 nm range were used. As shown in Figure 6b, obvious signals were observed after the probe was injected into mice for 5 min. However, with the increase in Na_2_SO_3_ concentration, the fluorescence signals gradually weakened (Figure 6c,d). As the response time is less than 5 min, it is very suitable for the real-time monitoring of SO_3_^2−^ in mice.

Based on the above results, we envisioned the mechanism of detection as follows (Figure 2). At the excitation wavelength of 380 nm, the donor (imidazo[1,5-a]pyridine) transfers energy to the acceptor (benzopyran) and NIR fluorescence emission at 760 nm was observed. However, after the addition of SO_3_^2−^, the reaction between SO_3_^2−^ and benzopyran breaks the π conjugate of benzopyran, resulting in the destruction of FRET, and thus, the energy of imidazo[1,5-a] pyridine cannot be transferred to the benzopyran. Therefore, the fluorescence emission at 760 nm disappeared and the emission at 475 nm increased. This is also confirmed by ^1^H NMR (Appendix A).

## 3. Experimental

### Synthesis of the Probe IPB-RL-1

As demonstrated in Figure 1, compounds **1–4** were synthesized according to the reported procedure [9,27].

Compound **3** (0.10 g, 0.24 mmol), compound **4** (0.10 g, 0.28 mmol) and CH_3_COOH (8 mL) were added to a 25 mL round-bottom flask. The mixture was heated to reflux for 3 h and then poured into water (100 mL). After being extracted with DCM (20 mL) three times, the combined organic solvent was removed under reduced pressure. The pure product was obtained by column chromatography (CH_2_Cl_2_:MeOH = 200:1). Black solid, ^1^H NMR (400 MHz, DMSO-*d*_6_) δ: 8.36 (s, 1H), 8.22 (d, *J* = 7.2 Hz, 1H), 8.00 (s, 1H), 7.78 (s, 1H), 7.59 (d, *J* = 8.8 Hz, 2H), 7.49 (s, 1H), 7.33 (dd, *J* = 9.2, 2.4 Hz, 1H), 7.19 (d, *J* = 2.4 Hz, 1H), 7.02 (d, *J* = 8.8 Hz, 2H), 6.70 (dd, *J* = 7.2, 1.6 Hz, 1H), 2.99–2.85 (m, 5H), 2.79 (s, 2H), 1.80 (m, 3H), 1.64 (m, 4H), 1.30 (m, 4H), 1.18 (m, 9H), 1.03–0.97 (m, 2H), 0.85 (m, 4H) ppm; ^13^C NMR (101 MHz, DMSO-*d_6_*) δ: 167.73, 164.09, 158.68, 155.65, 151.84, 138.94, 134.09, 132.07, 130.42, 128.61, 126.33, 125.51, 124.32, 123.74, 123.48, 122.83, 118.70, 118.15, 116.07, 114.57, 114.02, 112.81, 112.41, 45.83, 29.49, 29.12, 27.35, 25.65, 22.56, 22.20, 21.47, 14.42, 14.16, 13.02 ppm; HRMS: ([M]^+^) Calcd for C_40_H_45_ClN_5_O_2_: 622.3256; found: 622.3266.

## 4. Conclusions

In summary, a NIR ratiometric fluorescent probe **IPB-RL-1** with an ultra-large Stokes shift (460 nm) that is superior to most reported probes has been developed. **IPB-RL-1** shows high sensitivity and selectivity. Detection of SO_2_ in mitochondria in living SKOV-3 cells was also realized. Moreover, the probe was successfully used to detect SO_2_ in zebrafish which may be useful for the understanding of biological roles of SO_2_ in mitochondria and in vivo. However, due to the small overlap between donor emission and acceptor absorption of the probe **IPB-RL-1**, the fluorescence transfer efficiency is only 51%, which implies that in order to obtain a high fluorescence transfer efficiency, the overlap effect between donor emission and acceptor absorption, in addition to the distance between donor and acceptor, should be carefully considered for the FRET-based probe design in the future.

## Data Availability

The data that support the findings of this study are available from the corresponding author upon reasonable request.

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
