# Peer review of "An Imidazo[1,5-a]pyridine Benzopyrylium-Based NIR Fluorescent Probe with Ultra-Large Stokes Shifts for Monitoring SO2"

_molecules, 2023, doi:10.3390/molecules28020515_

Round 1

Reviewer 1 Report

The submitted manuscript reports the synthesis, characterisation and application of a fluorescent probe for SO2 (IPB-RL-1). The claimed novelty is that IPB-RL-1 has a very large Stokes shift, enabling fluorescence in the NIR. The manuscript is generally well written though it could be improved in some parts. The following points should be addressed before publication.

1) what is missing is a deeper analysis of the FRET mechanism, in terms of the structure of the molecule. Why with a short distance between donor and acceptor the FRET efficiency is only 50% (Fig. S6)? Is this due to not optimal overlap between the spectra or to orientation of the transition dipoles? Does this suggest routes to improve FRET efficiency? Given the long experience (seven papers about these systems by some of the same authors are cited in the supplementary materials) in realising SO2 fluorescent probes one would expect a more insightful and rational approach to design. 

2) The title does not mention the fact that the probe is  SO2 sensitive. What is a "NIR system"? The authors should find a more descriptive title.  

3) Though the authors make a good and useful summary of the existing SO2 probes in Tab S1, a recent work on the topic, entitled "Mitochondria targeted and immobilized ratiometric NIR fluorescent probe for investigating SO2 phytotoxicity in plant mitochondria" doi: 10.1016/j.snb.2022.132433 is not discussed. 

4) For readers not entirely familiar with the topic it might not be clear the connection between SO32- and SO2. Please clarify.

5) Figures have very large labels (for example "(a)" and "(b)" in fig 1 and 2) that overlap with the graphs. This is even a bigger problem in Figure 5 and 6.

Minor points (the numbers refer to the line)

18 shouldn't it be endogenous?

26 define ICT

31 its -> their

43-46 isn't the IPB-RL-1 probe the result of the previously described selection of donor and acceptor ? this should be better stated, as it looks like IPB-RL-1 is used in addition to the other two fluorophores.

94 what are sigma and k in the formula?

98 what is MTT?

98 lower cytotoxicity with respect to what?

101 was -> were

102-103 and 111 adjust the sequence of tenses. Also some other points in the manuscript need a better choice of the tenses. 

In Supplementary:

Define what the donor is in Fig. S5.

Fig. S6: how is this an "Analysis diagram of fluorescence transfer efficiency of IPB-RL-1."? How is the energy transfer efficiency obtained?  

Author Response

1) what is missing is a deeper analysis of the FRET mechanism, in terms of the structure of the molecule. Why with a short distance between donor and acceptor the FRET efficiency is only 50% (Fig. S6)? Is this due to not optimal overlap between the spectra or to orientation of the transition dipoles? Does this suggest routes to improve FRET efficiency? Given the long experience (seven papers about these systems by some of the same authors are cited in the supplementary materials) in realising SO2 fluorescent probes one would expect a more insightful and rational approach to design.

Response: Thank you very much for your suggestion. The following analysis has been added in the conclusion.

“However, due to the small overlap between donor emission and acceptor absorption of probe IPB-RL-1, the fluorescence transfer efficiency is only 51%, which implies that in order to obtain high fluorescence transfer efficiency, the overlap effect between donor emission and acceptor absorption in addition to the distance between donor and acceptor should be carefully considered for the FRET based probe design in the future.”

2) The title does not mention the fact that the probe is SO2 sensitive. What is a "NIR system"? The authors should find a more descriptive title. 

Response: We have modified the title as“Imidazo[1,5-a]pyridine-benzopyrylium based NIR fluorescent probe for monitoring SO2 with ultralarge Stokes shifts”

3) Though the authors make a good and useful summary of the existing SO2 probes in Tab S1, a recent work on the topic, entitled "Mitochondria targeted and immobilized ratiometric NIR fluorescent probe for investigating SO2 phytotoxicity in plant mitochondria" doi: 10.1016/j.snb.2022.132433 is not discussed.

Response: The reference has been added to Table S1.

[43] X.D. Chen, Q. Chen, D. He, S.X. Yang, Y.F. Yang, J. Qian, L.L. Long, K. Wang, Mitochondria targeted and immobilized ratiometric NIR fluorescent probe for investigating SO2 phytotoxicity in plant mitochondria. Sens Actuators, B. 370 (2022) 132433.

4) For readers not entirely familiar with the topic it might not be clear the connection between SO32- and SO2. Please clarify.

Response: We have added relevant content in the introduction section for readers to better understand the relationship between SO2 and SO32-.

“SO2 could dissolve easily in water to form its derivatives: bisulfite (HSO3-) and sulfite (SO32-). So the physiological functions of SO2 can be attributed to its derivatives (HSO3-/ SO32-).”

5) Figures have very large labels (for example "(a)" and "(b)" in fig 1 and 2) that overlap with the graphs. This is even a bigger problem in Figure 5 and 6.

Response: They have been modified accordingly.

6) shouldn't it be endogenous?

Response: The word has been revised accordingly.

7) define ICT

Response: It has been explained in the article.

“ICT (Intramiolecular Charge Transfer)”

8) its -> their

Response: The word has been revised accordingly.

9) 43-46 isn't the IPB-RL-1 probe the result of the previously described selection of donor and acceptor ? this should be better stated, as it looks like IPB-RL-1 is used in addition to the other two fluorophores.

Response: It has been revised in the introduction as follows:

“Owing to the good optical property[29], imidazole[1,5-a]pyridines were selected as the donor to construct the FRET platform [30]. In addition, we chose the benzopyran salt as the acceptor because of its NIR emission. Meanwhile, benzopyran moiety could not only be used as reactive site for Michael addition reaction with SO2 to achieve detection purposes, but also it could target mitochondria due to positive electricity. Therefore, the designed probe IPB-RL-1 could successfully achieve its imaging of SO2 in mitochondria in SKOV-3 cells.”

10) what are sigma and k in the formula?

Response: The explanations of sigma and k have been added.

“(LOD = 3 σ/ k, σ is the standard deviation of blank measurement, and k is the slope of fluorescence emission ratio (I475/I760) and SO32- concentration)”

11) 98 what is MTT? lower cytotoxicity with respect to what?

Response: All of them have been revised and marked.

“The results of MTT (Methyl Thiazolyl Tetrazolium) experiment (Fig.S4) showed that IPB-RL-1 has lower cytotoxicity to SKOV-3 cells and could be used for further cell imaging experiments.”

12) 101 was -> were

Response: The word has been revised accordingly.

13) 102-103 and 111 adjust the sequence of tenses. Also, some other points in the manuscript need a better choice of the tenses.

Response: All of them have been revised and marked.

14) Define what the donor is in Fig. S5.

Response: The donor is compound 3 and it has been added in the supporting information (Fig. S5).

15) Fig. S6: how is this an "Analysis diagram of fluorescence transfer efficiency of IPB-RL-1."? How is the energy transfer efficiency obtained? 

Response: The caption of Fig. S6 has been revised and the energy transfer efficiency has been explained.

Fig. S6 The emission spectrum of probe IPB-RL-1 and donor.

Energy transfer efficiency = 1-FDA/FD = 51%

Where FDA is the fluorescence intensity of the donor in the presence of the acceptor, FD is fluorescence intensity of the donor in the absence of the acceptor.”

Reviewer 2 Report

The paper by Qin and Ge is about a new fluorescent compound demonstrating fluorescence resonance energy transfer. The compound obtained has been used for imaging of SO2 in living SKOV-3 cells, zebrafish, and living mice. The paper is suitable for Molecules but needs minor revisions.

1.            The region above 400 nm is visible region, not UV. Moreover, all absorption spectra are UV-vis, not just only UV. Please correct within the whole manuscript.

2.            Practically all figures need careful revisions:

Figures 1-2 contain letters a, and b, but there are no references to them. These letters seem to be an extra.

Figures 5-6 contain a lot of unreadable letters.

3             The UV-vis and fluorescent experiments presented in figures 1-2 demonstrate the same intensities, but the authors use different concentrations (50 μM for fig 1, and 5 μM for 2). Please, Correct this.

4             Could a process of interaction with SO3 be reversible? Because it could be important for reusing the initial compound.

Author Response

1) The region above 400 nm is visible region, not UV. Moreover, all absorption spectra are UV-vis, not just only UV. Please correct within the whole manuscript.

Response: All of them have been revised and marked.

2) Practically all figures need careful revisions: Figures 1-2 contain letters a, and b, but there are no references to them. These letters seem to be an extra. Figures 5-6 contain a lot of unreadable letters.

Response: They have been revised accordingly.

3) The UV-vis and fluorescent experiments presented in figures 1-2 demonstrate the same intensities, but the authors use different concentrations (50 μM for fig 1, and 5 μM for 2). Please, Correct this.

Response: Thank you very much for your advice. We have corrected the mistakes.

4) Could a process of interaction with SO32- be reversible? Because it could be important for reusing the initial compound.

Response: In the process of the reaction with SO32-, HCHO/H2O2 could be adjusted to achieve the goal of reversible reaction, which provides the possibility for the follow-up research of probe recycling and various applications.